# Statistical Methods to Support Difficult Diagnoses

**DOI:** 10.3390/diagnostics11071300

**Published:** 2021-07-20

**Authors:** Guenter F. Pilz, Frank Weber, Werner G. Mueller, Juergen R. Schaefer

**Affiliations:** 1Institute of Algebra, Johannes Kepler University, 4040 Linz, Austria; 2German Air Force Center of Aerospace Medicine, 82256 Fuerstenfeldbruck, Germany; Dr.Frank.Weber@t-online.de; 3Institute for Applied Statistics, Johannes Kepler University, 4040 Linz, Austria; Werner.Mueller@jku.at; 4Center for Undiagnosed and Rare Diseases (ZusE), Philipps University, 35043 Marburg, Germany; juergen.schaefer@mailer.uni-marburg.de

**Keywords:** diagnosing designs, rare diseases, statistics, regression, block designs

## Abstract

Far too often, one meets patients who went for years or even decades from doctor to doctor without obtaining a valid diagnosis. This brings pain to millions of patients and their families, not to speak of the enormous costs. Often patients cannot tell precisely enough which factors (or combinations thereof) trigger their problems. If conventional methods fail, we propose the use of statistics and algebra to provide doctors much more useful inputs from patients. We use statistical regression for triggering factors of medical problems, and in particular, “balanced incomplete block designs” for factors detection. These methods can supply doctors with much more valuable inputs and can also find combinations of multiple factors through very few tests. In order to show that these methods do work, we briefly describe a case in which these methods helped to solve a 60-year-old problem in a patient and provide some more examples where these methods might be particularly useful. As a conclusion, while regression is used in clinical medicine, it seems to be widely unknown in diagnosing. Statistics and algebra can save the health systems much money, as well as the patients a lot of pain.

## 1. Introduction

In medicine, a diagnosis of a problem of a patient is usually generated by medical knowledge and experience, often using results of labs and other tests. The success rate for correct diagnoses is high if the inputs tell a clear message, such as in case of a broken bone. In other cases, however, such as for heavy headache, extreme weakness, etc., the situation is not so simple, and might require a much deeper search. Often enough, a satisfactory diagnosis is not found.

In fact, the number of patients without a valid and correct diagnosis is frighteningly high in areas where a diagnosis is non-trivial, e.g., in cases of rare diseases, if there is a huge number of possible triggers, or if decisive parameters are hardly measurable (such as in stress). A center for rare diseases in Germany presently has a backlog of more than 9500 desperate requests; a quick and informal search among an organized group of patients for a special rare disease revealed that more than 85% of them had no valid diagnosis.

A rare disease is defined by a prevalence of not more than 1 to 2000 inhabitants (https://www.orpha.net/consor/cgi-bin/Education_AboutRareDiseases.php?lng=EN, accessed on 17 July 2021). However, due to the fact that there is an estimated number of more than 8000 different rare diseases, the total number of patients with rare diseases is rather high. Thus, one might estimate that more than 300 million people on earth suffer from a rare disease. Even more patients are afflicted with “incomplete” diagnoses due to hardly measurable or subjective (but wrong) inputs of patients.

This dramatic situation might be improved by an increasingly expensive medical machinery, but also by the use of statistical regression, which tells patients (and their doctors) much more about their triggering factors than they are aware of. Surprisingly little has been done thus far in this direction, except in clinical research. Two rather new books (see [1,2]) provide first systematic accounts on regression in medicine, but with no emphasis on diagnosing, and block designs for dependent factors are not covered in it at all.

Here are some examples where a traditional medical search might be too slow, too complicated, or too expensive, but where mathematics and statistics can provide reliable results is a very short time:Some (especially elderly) people often take a large number of drugs. Often enough, some of these drugs (or combinations of them) can be the reason for further severe problems. Yet many of these interactions are not known well enough. Simply think that for the 1000 most frequent drugs, there are half a million possible interactions, which also differ from patient to patient. Statistics provides an exceptional tool to test many of these interactions at the same time, using sophisticated mathematical methods, such as block designs and matrix calculus. Note that it is useless to find drug interactions for a large number of patients—they have an individual character. For instance, “dizziness” is on almost all package inserts, and hence is useless. We describe this method in more detail in Section 3.1 below. Note that this application does not concern rare diseases, but remarkably frequent cases in treating patients.Reactions to the intake of food (components) can provide valuable hints for the diagnosis. However, one cannot expect patients to know that, for instance, they react to an imbalance of magnesium intake. In the section “Statistics Works” below, we describe a case where problems through the intake of too much potassium but too little sodium caused severe and frequents attacks of paralysis for more than 50 years. This led doctors to investigate a gene that was previously not considered as a cause of paralysis (see [3]), as well as to find the defect. The solution of this case and the fact that that this defect seems to be unique worldwide shows the power of the statistical approach. This case was described in detail in [4].Combinations of allergens can be tricky. The authors of [5] describe cases in which one allergen is neutral for the patient, another one positive, but the combination is a complete disaster. Our methods can detect cases such as this without problems.

The situation is intensified by the fact that a small change in the input might result in a large change of the output (= diagnosis), no matter whether the search for the diagnosis is computer-aided or not. In mathematical language, the output does not depend continuously on the input. Hence, in crucial situations, it might be highly desirable to improve the quality of the inputs. The statistical approach usually does need the assistance of a statistician (in the near future maybe simplified by an app) and the cooperation of the patient, but nevertheless it is far less expensive than a complicated medical machinery, or a wrong diagnosis.

## 2. Materials and Methods

### 2.1. Statistical Methods I: Regression Analysis

The role of statistics in life sciences is ubiquitous—simply think of the millions of statistical tests for the efficiency of medications or medical treatments, or trials on (sometimes many thousands of) patients (see, e.g., Cleophas et al. [1]). Less common is the use of statistics to identify one or more of a large number of factors that might trigger pain or discomfort in a single patient (“precision medicine”); an account was only given recently by Cleophas and colleagues [2]. Very rarely, a search is made to find positive or negative synergy effects (interactions) between these factors that go far beyond a mere addition of these factors. The reason for that is, of course, the huge number of possible combinations of two or more factors. For the sake of the patients, however, the number of tests should be as small as possible. We present a solution to this dilemma. The identification of these “suspicious” factors can be very valuable in obtaining a diagnosis when this turns out to be difficult.

Our statistical method, as already briefly described, is that of (statistical) regression. First, the patient and the doctor together try to find out which parameters (“factors”, say *x_1_, x_2_, …*) might improve or worsen the patient’s situation. These must be in some way “measurable”, and they also must find a numerical indicator *y* that describes the patient’s situation. The “degree of stress” or “pain on a scale from 0 to 10” are acceptable, but parameters such as “blood pressure” would be better, of course. The number of parameters should not be “too low” (danger of missing the most useful parameters) and not “too large” (resulting in a huge number of tests); usually, a size between 5 and 15 might do the job. Of course, if the situation of the patient stays at the same level all the time, statistics is of no use.

Then, a statistician selects an efficient experimental design, explaining which factors (such as pills or food components) the patient should try on, say, day 1, which factors should be tested on day 2, etc. The patient notes the resulting state *y* of his situation after each test. The “protocol” might then look like
Day 1: I tried *x_1_, x_3_, x_4_, x_7_,* the result was 23;Day 2: I tried *x_2_, x_3_, x_5_, x_9_,* the result was 19;
and so on. Usually, one wants a design in which all factors are tested an approximately equal number of times. In statistics, this is usually called a “screening experiment”. 

The patient sends this protocol to the statistician (or doctor). The statistician, using regression, finds those factors (or combinations thereof) that are very likely to improve the patient’s situation, and the “bad” factors that worsen it. Irrelevant factors are detected automatically.

In medical statistics, regression analysis is usually used to analyze large samples, e.g., stroke risk as a function of age, hypertension, smoking habits, etc. Up to now, the use to find a diagnosis, however, is very rare. Here, we show that statistical regression can be very useful in the diagnosis of an individual case by detecting unknown connections between a number of “suspicious” factors.

Of course, this method is much more conspicuous than a usual diagnosis, and so it will only be used in cases where conventional methods have failed. However, doctor’s waiting rooms often contain patients who have run through an unsuccessful series of many tests generating numerous diagnoses. This can be very frustrating and sometimes also dangerous for them, usually taking much longer than the method we are demonstrating here. In particular, the diagnostic path in patients with rare diseases may be troublesome. Often, the patient can undertake the tests and measure the results by themselves.

The method used can perhaps be seen best via an example. Take a patient with unknown factors that trigger an allergy, wherein the usual diagnostic measures did not yield a satisfactory result. Suppose that the patient and the doctor suspect that n more factors *x_1_, x_2_,…, x_n_* might explain the allergy, e.g.,
*x_1_* = exhaust air of the vacuum cleaner (measured in minutes of exposure);*x_2_* = intake of certain candies (measured in pieces), …
and so on. Then a test plan might ask the patient for an exposure to *x_1_, x_4_,* and *x_9_,* and to rank the degree *y_1_* of allergy on, say, a scale of 10 degrees of severeness. The second test might involve *x_2_, x_4_,* and *x_6_,* with a result of *y_2_*, and so on.

Then, well-known statistical algorithms (see, e.g., Morris [6]) will yield a “formula” of the type
*y = β_0_ + β_1_x_1_ + β_2_x_2_ + … + β_n_x_n_*(*)
where *β_0_* is a constant (the “intercept”) and *β_i_* estimates the “true” influence b_i_ of *x_i_* to the overall allergy level. Usually, one also determines confidence intervals [*β_i_ − c_i_, β_i_ + c_i_*] so that they cover b_i_ with a confidence level of, say, 95%. If this interval covers 0, such as for example in [−0.4, 0.7], one usually reacts in the way that the influence of the corresponding factor *x_i_* is to be doubted (not statistically significant) and thus *x_i_* is eliminated from the list of interesting factors. This usually happens for many factors, such that eventually a small list of suspicious factors remains, and the doctors will pay their attention to these few factors. This reduction is often essential, because it makes a huge difference whether 100 or 3 factors have to be medically investigated. The patient might already be dead when the doctors come to explore factor # 50… 

If the tests conducted do not yet give results which are significant enough, one should continue the tests (for instance, by repetition). Then, the software will “learn” more and more about the case of investigation, which is the principal of artificial intelligence (AI).

### 2.2. Statistical Methods II: Experimental Designs

Up to now, we applied a known algorithm to a rather new situation. Things become less simple if the important factors cannot be accounted for additively, but interactions (“synergy effects”) are possible. Then, one usually adds terms such as *x_i_x_j_* to *x_1_,x_2_,…,x_n_* in the analyzed model. In medicine and biology, often two substances (substance and antidote, …) work together to produce an effect. For more examples, see below.

However, much more care must then be taken to the design of the experiments. It might be that *x_i_* and *x_j_* are never (or only once) tested together, and therefore no clarification of a synergy is possible. 

The fairest way would be to test every *x_i_* the same number of times, and also to test every pair *x_i_* and *x_j_* the same number of times. A new problem now comes from the quick rise of binomial coefficients. With 5 factors, we have 10 possible pairs, but with 20 factors, we already have 190 pairs. Thus, a clever trick is needed: we utilize some particular experimental designs:

**Definition:** A *Balanced Incomplete Block Design* (*BIB-design*), see, e.g., Lidl and Pilz [7], consists of a set *P = {p_1_, p_2_, … , p_v_}* of *v* “points” and a collection *B* of *b* subsets *B_1_, B_2_, … , B_b_* of *P* (called “blocks”), such that
(i)Each point in *P* belongs to the same number *r* of blocks;(ii)Each *B_i_* has the same number *k* of elements;(iii)Each pair *p_i_, p_j_* of points belongs to the same number *λ* of blocks.The pair *(P, B)* is then called a *(v,b,r,k,λ)-**design*. The design is *complete* if *B* is just the collection of all *k*-element subsets, otherwise *incomplete*.For an experiment such as the one above (concerning allergies), a BIB-design can be turned into an experimental design as follows. 
The points are the factors (e.g., possible triggers for an allergy);Every block lists the factors which will be tested simultaneously in a test.
Thus, a *(v,b,r,k,λ)*-design will test *v* suspected triggering factors; each test requires *k* suspected factor (at the same time), and one will need *b* tests. Number (i) above assures that every possible triggering factor will be tested the same number (namely, *r*) of times, and every pair of possible factors will be tested together in exactly *λ* tests. Therefore, a BIB-design provides an experiment that is “fair”, both to the factors and the tests. It is not trivial at all to obtain such a design. Constructions usually come from areas “far away”, such as from finite geometries or abstract algebra (structures such as groups or near-rings).

**Example:** From mathematical considerations (see Ke-Pilz [8]), we might obtain the following design (which comes “out of the blue” now, but we will not provide the long mathematical derivations):
*P =* {1, 2, 3, 4, 5, 6, 7} and *B* consist of the 14 collections*B_1_ =* {2,4,5}, *B_2_ =* {1,3,7}, *B_3_ =* {1,2,6}, *B_4_ =* {1,5,7}, *B_5_ =* {1,3,4}, *B_6_ =* {2,3,7}, *B_7_ =* {4,5,7},*B_8_ =* {1,2,4}, *B_9_ =* {2,6,7}, *B_10_ =* {2,3,5}, *B_11_ =* {3,4,6}, *B_12_ =* {3,5,6}, *B_13_ =* {1,5,6}, *B_14_ =* {4,6,7}

This produces a (7,14,6,3,2)-design. Suppose we have 7 factors *x_1_, x_2_, … , x_7_*. For the first test, we try the factors *x_2_,x_4_*, and *x_5_*, since *B_1_ =* {2,4,5}, and so on (Table 1):

Thus, we have *v =* 14 tests, plus a “zero test” for “technical reasons” (the information matrix would otherwise not have full rank). One sees:Every test (except #15) involves *b = 3* factors (3 dots in every column);Each factor is tested in *r = 6* tests (6 dots in each row);Each pair of factors is tested together in *λ = 2* tests.

Observe that we tested the 21 possible pairs *x_i_* and *x_j_* twice in only 15 rather than 2 × 21 = 42 tests. This “magic reduction” to only one-third of the tests can be attributed to the fact that in each test, three synergies are considered simultaneously (in terms of experimental designs: we allow certain interactions to be aliased).

In the last row, we have supplemented some (fictitious) results of the tests. Linear regression provides the best estimates according to (*) in the section “Statistical Methods I” as
*y = 3 + 51x_4_ + 19x_5_ − 41x_6_*(Model 1)

If one also uses interaction terms (“synergies”), one instead obtains
*y = 2 + 47x_4_ − 31x_6_ + 58x_2_x_5_*(Model 2)

Now we can compare the actual results with the predicted ones using these two models (see Table 2):

One easily sees that Model 2 describes “the reality” considerably better than Model 1.

Let us remark that estimating possible product terms creates a problem because of the very small number of tests. We first look at the “main effects” *x_i_*, remove the irrelevant ones, and always add one if the *x_i_x_j_* to check which of them seemed to be statistically relevant. These are then added to the relevant main effects (thereby following the so-called hereditary principle). The final result might depend on these choices and their ordering but performing the calculations several times with different choices and orderings may yield a robust choice. Moreover, for patients, it does make a large difference if they have to undergo many more tests. Of course, repetitions of these 15 tests would also provide much sharper results.

## 3. Results

### 3.1. Statistic Works

We herein use the described method in “real” situations. First, as already addressed in the introduction, we describe the search for a diagnosis in a patient. His problems started at the age of 15 years with unusual tiredness attacks, always in the late afternoon. In the following 50 years, the tiredness worsened to complete paralysis attacks, in which the patient was fully conscious, but could not move any part of his body. There was no way to communicate with the world around him. Despite the consultation of more than 120 doctors and hospitals over decades, no reason was found. Since the patient reported an association of the paralysis attacks with a low intake of carbohydrates, the doctors first thought that a low blood sugar level might be the reason. This did not turn out to be true in this form, and thus doctors specialized for rare diseases considered the possibility of a periodic paralysis due to a defect in ion channels. However, the ion levels within the blood were normal during the attacks, which is typical in patients suffering on normokalemic periodic paralysis. Thus, it was tricky to identify the intracellular mechanism, namely, too high or too low levels of sodium or potassium, respectively. The doctors mentioned that it might take a long time to check all these channels.

The patient (a mathematician) tried to support the research team by speeding up the search process. He noted each of his meals and how much calcium, sodium, potassium, protein, etc. he had eaten. He checked his body strength by pressing a bathroom scale, right after the meals, and again one hour later. The differences came up to about ±10 kg. Thus, he first used statistics (Model 1) above to identify the few most probable intake components that explain the differences, using the software package *Mathematica^®^*. After a period of about 5 weeks of carefully documented eating, gained had the result
*y = y(p,s) = −0.5 − 0.0048p + 0.0085s*
with the interpretation that 1 h after the intake of *p* mg of potassium and *s* mg of sodium, the patient could press the scale (on average) *y(p,s)* kg harder. The signs of the coefficients for *p* and *s* were significant at the 99% level, which means that potassium hurts the patient, while sodium helps him. The exact values of these coefficients are not so important, except that the patient now knows in advance that, for example, a typical burger will strengthen him by approximately *y*(420,1000) = 6.0 kg, while 100 g of bananas will weaken him by *y*(390,1) = −2.4 kg.

Then the patient ran another test, this time with a BIB-design similar to the example above to check if possibly a combination of other substances might overthrow this result. However, no relevant combination was found, and therefore the above result was accepted. In this case, however, no test strictly according to the BIB-plan was possible, since no food contains only potassium, calcium, and sodium, and no other substances. We will return to this point later.

After this finding, the doctors knew that they had to search for a defect in a potassium channel and, more importantly, that lowering potassium should be beneficial in this special patient. The subsequent analysis of various ion channels revealed a thus far unreported gain of function by increased expression of the inward rectifier potassium channel Kir2.6, due to highly increased promotor activity of the gene KCNJ18. Interestingly enough, this gene was considered thus far as a less likely candidate for paralysis and was no target candidate in well-established screening panels (see Kuhn and colleagues [3]). The study and results were reported recently elsewhere by Soufi and colleagues [4]. Note that without the doctor’s hint to consider ion channels, the patient would never have conducted these experiments. Moreover, without medical competence, the results of the experiments could not have been properly interpreted. Thus, this case might be considered as a fine and successful interplay between medicine, statistics, and abstract algebra.

### 3.2. More Examples

(1)**Side effects of combinations of drugs:** This was also mentioned in the introduction. We had a case of a person (aged 75) who developed a strong and permanent dizziness that did not allow him to drive a car any more. He took 10 types of drugs per day, and we added the consumption of a standardized amount of alcohol as “drug # 11”. Therefore, we used the following (11,22,5,10,4)-block design as in Table 1 and added another “test”, this time”: the usual drug consumption of the patient. Table 3 shows:

Thus, in the first test, we gave drugs no. 1, 4, 7, 8, 9, and so on. If we exclude “test” 23, every drug was tested 10 times, and each pair of drugs was tested four times together. If one used single tests, we would have needed 10 × 11 + 4 × 55 = 330 tests. If the physicians (and not the statistician) said that each test needs 1, 2, or 3 days (depending on the half-life of the drugs, etc.), this program needs 22, 33, or 44 weeks. With single tests, this would be almost 1, 2, or 3 years. No patient would agree (“a patient is often impatient”).

Let us mention that such a design greatly reduces placebo and nocebo effects (which are typical and critical for tests with single medications), since the patients will be “confused” by the relatively large numbers of drugs prescribed/not prescribed per day.

Here, we found that we could exclude (on the 95% level) any side effect coming from the drugs. This is much more than to say, “we did not find any side effect”.

(1)**Food-dependent, exercise-induced anaphylaxis:** The contact with some allergens might be harmless, and physical exercise can help a lot, while the combination can be disastrous. Thus, one factor is neutral for the patient, the other one positive, but the combination is really negative (see Romano et al. [5]).(2)**Phototoxic dermatosis:** We usually tolerate sunlight at a usual dose very well since it is essential for our survival. Frequently used medications such as certain antibiotics, non-steroidal anti-inflammatory drugs (NSAIDs), and diuretic and antiarrhythmic drugs have usually no direct side effects at the skin. However, these drugs are known to enhance photosensitivity. In combination with usually well-tolerated sunlight, these drugs can create severe sunburn such as skin reactions (see [9,10]).(3)**Hyperkalemic periodic paralysis:** In a mild form, this disease can usually be tolerated, but in combination with a pathogenic gene mutation, it can create severe paralysis.(4)**Stomach problems:** A patient complains about stomach pains after some meals. His doctor suspects that seafood might be a reason, but this can hardly explain the pains. Moreover, he can exclude a large number of food components that do not hurt the patient. However, 15 “suspicious” factors remain. The following might be a typical progression of the statistical investigation. A simple regression test as in “Statistical Methods I” quickly excludes eight of them. For the remaining seven components, this test does not give satisfactory results. Therefore, we might use the test in “Statistical Methods II”. Suppose that the remaining seven factors are sugar (= S), apples (= A), lactose (= L), walnut (= W), pepper (= P), crabs (= C), and mustard (= M). So, according to the experimental design in “Statistical Methods II”, the first test would be a meal with S, A, and W. Then, a statistician quickly finds out that C does hurt a bit (as single factor), but the combination A and P is the main reason for the pains, while A and P alone do not really hurt.

## 4. Discussion

Many statisticians might be unhappy with several parts of the statistics used above. Metric and ranked data are mixed, the number of tests (especially in Model 2) can be dangerously low, the BIB-plan above should be filled with 0–1-data and not with real numbers, the ranking of pain by patients is highly subjective, and so on. 

An important point concerns the kinds of “dependences” involved. Dependent factors for a regression can be treated using BIB-designs, as we have seen. The tests suggested are by no means independent of the particular patient (see the next paragraph), as we are making no patient-to-patient comparisons. We do not aim at general results when they do not exist. Moreover, there might be a time dependence between the tests. For mastering that, physicians (and not statisticians) have to decide on the optimal time distance between the tests in order to guarantee independent observations as we assumed. Should one want to relax those assumptions, one might have to employ more complex design strategies, such as, e.g., those given in Kiefer and Wynn, [11]. Moreover, almost all statistical investigations on patients lack an important feature: they are not reproducible, such as statistics in technical sciences. See, for instance, the brilliant article by Homes [12]. 

However, medicine is not pure natural science, and it might be better to use a partly “dirty” statistics than to do nothing. In addition, most of all, the statistical results are not the diagnosis, but simply suggestions providing process for an appropriate medical investigation. Still, the medical part of obtaining the diagnosis is by far the most important one. However, as can be seen in our case report, statistics can be very helpful. 

In fact, the statistician plays an important role: they have to identify, together with the doctors and the patient, which of the hundreds of possible factors might be relevant. A careful selection is necessary. Hence, the statistician must be proficient in “model building”. Let us note that the statistical models mentioned above are also useful (and have been employed) in many other areas, cf. the survey on spatial applications by Mueller [13]. In agriculture, the factors *x_i_* might be fertilizers, irrigation, etc.; in paint manufacturing, they might be additives against weather attacks, and so on. 

## 5. Conclusions

We present a seemingly unknown and inexpensive tool for obtaining valid diagnoses in difficult cases, especially for rare diseases, but also for everyday problems. The basic idea is to employ statistical regression in order to obtain much more precise inputs from patients. They often do not know exactly which substances, actions, and circumstances (or combinations thereof) trigger their problems. Statistics can often easily explain which of these factors contribute to the worsening of the patient’s problems. We found that this precise information often leads the way to the correct diagnosis.

Typically, the patients can gather the necessary data by themselves, in measuring parameters such as blood pressure, intake of food and drugs, physical strength, degrees of the pains, and so on. We plan to develop an app to facilitate the collection for the patients. Since more and more medical information will be gathered be so-called “wearable sensors”, we can expect a rapid increase of data. These data must be well organized to be useful for the physicians.

There is a vast amount of literature on the use of statistics in medicine, wherein the data are collected by patients (see, e.g., Saunders and colleagues [14]), but they have a completely different approach. 

Therefore, we believe that the use of statistics can help physicians to resolve difficult cases, for instance, in cases of rare diseases when the usual methods seem to fail. This, for sure, will greatly relieve many deeply unhappy people among the large number of undiagnosed patients.

## Figures and Tables

**Table 1 diagnostics-11-01300-t001:** An experimental design for testing 7 factors, each of them 6 times.

Fact.\Test	1	2	3	4	5	6	7	8	9	10	11	12	13	14	15
***x_1_***	●	●	●	●	●								●		
***x_2_***	●		●			●		●	●	●					
***x_3_***		●			●	●				●	●	●			
***x_4_***	●				●		●	●			●			●	
***x_5_***				●			●	●		●		●	●		
***x_6_***			●						●		●	●	●	●	
***x_7_***		●		●		●	●		●					●	
**Results**	**49**	**−2**	**−28**	**3**	**54**	**−1**	**51**	**98**	**−31**	**69**	**18**	**−35**	**−25**	**22**	**3**

**Table 2 diagnostics-11-01300-t002:** Comparing two test results with reality.

Test	1	2	3	4	5	6	7	8	9	10	11	12	13	14	15
real	**49**	**−2**	**−28**	**3**	**54**	**−1**	**51**	**98**	**−31**	**69**	**18**	**−35**	**−25**	**22**	**3**
Mod.1	54	3	−38	22	54	3	73	73	−38	22	13	−19	−19	13	3
Mod.2	49	2	−29	2	49	2	49	107	−29	60	18	−29	−29	18	2

**Table 3 diagnostics-11-01300-t003:** *A* (11,22,5,10,4)—block design.

1	2	3	4	5	6	7	8	9	10	11	12	13	14	15	16	17	18	19	20	21	22	23
●	●			●	●					●	●					●	●	●	●			●
			●		●	●		●		●		●			●		●	●		●		●
			●	●			●	●		●			●	●				●	●		●	●
●		●	●					●	●		●	●	●				●		●			●
	●	●								●		●	●		●	●			●	●	●	●
	●	●	●	●	●	●	●				●	●									●	●
	●					●			●		●		●	●			●	●		●	●	●
●		●				●	●	●		●	●			●		●				●		●
●	●		●		●		●		●					●	●				●	●		●
●				●		●	●		●			●	●		●	●		●				●
		●		●	●			●	●					●	●	●	●				●	●

## Data Availability

There is no restriction on data. The datasets used and/or analyzed during the current study are available from the corresponding author on reasonable request.

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
