# Peer review of "Statistical Methods to Support Difficult Diagnoses"

_diagnostics, 2021, doi:10.3390/diagnostics11071300_

Round 1
Reviewer 1 Report
The authors took into account the suggestions/reviews of the first submission, improving the quality of the submitted article
Author Response
Please see the file attached.

Reviewer 2 Report
The manuscript "Statistical methods to support difficult diagnoses" by Pilz et al has presented an interesting method to help support the disease diagnosis which could be of great importance.
The authors need to improvise on the introduction part. if possible, they should provide more worldwide pictures rather than focusing on European data.
There are few typographic errors such as "prevalence of ≤ 1 to 2.000 inhabitants" at line 25 and line 27. It appears like 2 rather than 2,000.
Authors should also cite literature where similar or other statistical methods have been used for such cause.
Lastly looking at the manuscript it appears like it has been revised but I was not provided with any review report. Please avoid this if it is not the case.
Author Response
JOHANNES KEPLER
UNIVERSITY LINZ
Altenberger Str. 69
4040 Linz, Austria
www.jku.at
DVR 0093696
Linz, July 14, 2021
2nd revision – cover letter
Ref.: diagnostics-1203254
Dear Editors,
Dear Reviewers,
we want to thank our reviewers for their comments, which helped
us to improve the quality of our paper. We have revised our paper
entitled “Statistical Methods to Support Difficult Diagnoses” ,
according to the comments of reviewer 2.
Reviewer 2 has mentioned that (s)he has not received our first
revision. Reviewer 2 and reviewer 1 had similar comments (which
indicates the high quality of your reviewers), so we think that most
of the changes the second reviewer has asked for are already
incorporated in revision 1. Hence we worked in only the additional
suggestions of reviewer 2. This reviewer should definitely get our
first revision (this is the one presently listed on your website for
our paper).
Prof. emer. Dr. Dr. h. c.
Günter PILZ
Institute for Algebra
P +43 732 2468 6861
F +43 732 2468 6852
guenter.pilz@jku.at
Office:
Monika Peterseil
Ext. 6850
Monika.Peterseil@jku.at
To the Editors of
“Diagnostics”
JOHANNES KEPLER
UNIVERSITY LINZ
Altenberger Str. 69
4040 Linz, Austria
www.jku.at
DVR 0093696
Here are our responses to the comments of reviewer 2:
(1) Dear reviewer, we think that most points you addressed
in your Review Report Form have been addressed also
by reviewer 1, and we have changed our manuscript
accordingly already in our first revision.
(2) We provided worldwide figures rather than European
ones.
(3) We are very sorry that we caused a misunderstanding:
In Europe, we write 2.000 for “two thousand”, while in
English it would be 2,000. We have corrected that.
(4) There is simply no literature where similar or other
statistical methods have been used for similar cases
we described in our paper. We mentioned this in
Section 5 (“Conclusion”).
We hope our paper now fulfills all requirements for publication in
“Diagnostics” and thank you, and especially our reviewers, for
your support to improve the quality of our paper. Please let me
know it if you need more information.
With kindest regards
(Prof. emer. Günter Pilz)
Corresponding author
